# Molecular alterations in the integrated diagnosis of pediatric glial and glioneuronal tumors: A single center experience

**Sandra Lorena Colli[1], Nazarena Cardoso[1,2], Carla Antonella Massone[2], María Cores[3], Mercedes García Lombardi[3], Elena Noemí De Matteo[1,2], Mario Alejandro Lorenzetti[2], María Victoria Preciado**[2]*

**1** División Patología, Hospital de Niños "Dr. Ricardo Gutiérrez", Buenos Aires, Argentina, **2** Laboratorio de Biología Molecular, División Patología, Instituto Multidisciplinario de Investigaciones en Patologías Pediátricas (IMIPP), CONICET-GCBA, Hospital de Niños "Dr. Ricardo Gutiérrez", Buenos Aires, Argentina, **3** Unidad de Oncología, Hospital de Niños "Dr. Ricardo Gutiérrez", Buenos Aires, Argentina

* preciado@conicet.gov.ar

**Data Availability Statement:** All relevant data are within the paper and its Supporting Information files.

## Abstract

Objectives: Tumors of the central nervous system (CNS) are the most common pediatric solid tumors, where low grade (LGG) and high grade gliomas (HGG) represent up to 55% of CNS tumors. Current molecular classification of these tumors results in a more accurate diagnosis and risk stratification, which ultimately enables individualized treatment strategies. Identifying known alterations is a suitable approach, particularly in developing countries, where NGS approaches are not easily accessible. We sought to assess molecular alterations in BRAF and histone 3 genes. Study design: FISH, IHC and Sanger sequencing were performed in a series of 102 pediatric glial and glioneuronal tumors. We also correlated these results with clinical and histological findings to evaluate their usefulness as diagnostic and/or prognostic tools. Results: We found that the KIAA1549-BRAF gene fusion was a relevant diagnostic tool for pilocytic astrocytoma, but not related to progression free survival (PFS) and overall survival (OS). BRAFV600E mutation was associated with a decreased OS in LGG, and with decreased PFS and OS among pilocytic astrocytomas. All HGG of the midline were H3K27M mutants, while H3G34R mutant cases were located in brain hemispheres. HGG harboring the H3K27M variant were associated with a decreased PFS and OS. Conclusions: Assessing druggable molecular markers with prognostic value is particularly important in those cases where complete resection or further radiation therapy is not possible. These potential diagnostic/prognostic markers may be suitable as further screening tests to reduce the requirement on NGS, which is not available in all laboratories. Furthermore, these results broaden data on BRAF and Histone 3 alterations in children from geographic regions, other than USA and Europe.

**Funding:** This study was supported in part by a grant from the National Institute of Cancer (INC) of the National Ministry of Health (Res 83/2020) and National Research Council (CONICET) PUE 2018 n° 0058. E.N.D.M, M.A.L, and M.V.P are members of the CONICET Research Career Program. C.A.M. was supported by a fellowship from INC and N.C. by a fellowship from Buenos Aires City Ministry of Health. There was no additional external funding received for this study. The funders had no role in study design, data collection and analysis, decision to publish, or preparation of the manuscript.

**Competing interests:** The authors have declared that no competing interests exist.

## Introduction

Tumors of the central nervous system (CNS) are the most common solid tumors in pediatric patients; all together they account for more than 20% of all childhood cancers [1]. Within CNS tumors, gliomas represent up to 55% of them. The World Health Organization (WHO) defines these pediatric brain tumors into four stages based on their morphology and histological characteristics, where grades I and II (GI and GII) are considered low grade gliomas (LGG) and grades III and IV (GIII and GIV) are considered high grade gliomas (HGH) [2]. This classification reflects the survival odds and generally speaking, grade I have the highest survival odds and grade IV, the lowest. Pediatric LGG include pilocytic astrocytoma (GI), pleomorphic xanthoastrocytoma (GII), diffuse astrocytoma (GII), those entities related to ganglioglioma (GI/II) and dysembryoplastic neuroepithelial tumor (GI). Pediatric HGG are a heterogeneous group of malignant and poorly delimited tumors which extend both macroscopically and radiologically, beyond apparent margins. The most frequent in pediatrics include anaplastic astrocytoma (GIII), glioblastoma (GIV), diffuse intrinsic pontine glioma (GIV) and diffuse midline glioma H3 K27M-mutant (DMG K27M) (GIV).

According to the Argentine Oncopediatric Hospital Registry (ROHA, *Registro Oncopediátrico Hospitalario Argentino*), approximately 1400 new cases of cancer are diagnosed annually in our country in children under 15 years of age, where CNS tumors are the most frequent solid tumors [3]. Incidence rates in Argentina are similar to those described in other regions [1]. However, in recent years and given the astonishing advances in molecular biology and its implications in diagnosis, prognostic accuracy and tailored-made treatment strategies of pediatric brain tumors, specific molecular markers for each tumor type have been transferred from the bench to the bed-side [4]. More precisely, DMG K27M was defined by WHO based on the molecular characterization of the *H3F3A*, *HIST1H3B* and *HIST1H3C* genes [2]. This entity harbors a H3K27M hallmark somatic mutations in the histone 3 (H3) isoforms H3.3 (coded by *H3F3A* gene) or isoform H3.1 (*HIST1H3B* or *HIST1H3C* genes) in up to 80% of pediatric diffuse midline gliomas, including thalamic glioma and diffuse intrinsic pontine glioma (DIPG) [5]. A different hallmark mutation, H3G34R/V has been described within the same histone genes in gliomas arising from the cerebral hemispheres [6]. These variants have become the most frequently studied in gliomas; more precisely, H3K27M directly disturbs the target site for lysine specific methyltransferases and prevents methylation of the mutated residues, where the original lysine would have been methylated. Additionally, H3G34R/V interferes with the modification of the nearby K36 residue [7]. Given that H3K27M mutant gliomas have a decreased treatment response and a diminished overall survival, in relation to the non-mutant tumors, H3K27M is now considered a tumor driver mutation and is used as a prognostic marker.

In a similar fashion, the V600E mutation in BRAF (coded by the *BRAF* gene) highly correlates with a poorer prognosis and overall survival across a broad spectrum of pediatric LGG, while the *KIAA1549-BRAF* gene fusion aids in the diagnosis of pilocytic astrocytomas (PA) [4]. The MAPK signaling pathway regulates various cellular functions, including cell cycle control, cell proliferation, differentiation and apoptosis, in all of which BRAF acts as a stimulatory factor [8]. Moreover, BRAF V600E mutation has been reported to abnormally up-regulate the MAPK signaling pathway [9, 10]. Even though the BRAF V600E mutation responds unfavorably to conventional treatment, it represents a specific druggable target for personalized therapies with kinase inhibitors [11]. Moreover, *CDKN2A* has been previously shown to influence the response to treatment and outcome in BRAF V600E pediatric LGG [12–14]. On the other hand, the *KIAA1549-BRAF* gene fusion is formed by a tandem duplication of a 2 MB region (locus 7q34 in *BRAF* gene) with the subsequent fusion to *KIAA1549* gene, which finally results

in the constitutive phosphorylation activity by BRAF and up-regulation of the MAPK pathway [15]. Although several translocation break-points were described, the most common is between exon16 of the *KIAA1549* gene and exon 9 of *BRAF* (16–9 fusion); for a comprehensive review refer to [16]. While the BRAF V600E mutation is found in about 60% of pleomorphic xanthoastrocytoma (PXA), 50% of gangliomas, 9% of PA and 2–12% glioblastoma, the KIAA1549–BRAF fusion is frequently found roughly in 50–80% of PA and pilomyxoid astrocytomas (PMA), which is an aggressive variant of PA [16]. Despite all these facts, *BRAF* molecular profiling has not yet been included in the latest WHO classification of central nervous system tumors.

The molecular assessment of these markers aids in diagnosis, in identifying risk groups and individualizing treatment strategies, a fact of special importance in pediatrics since it may avoid unnecessary sequelae in a developing child. This approach, assessing known alterations by standard molecular techniques is useful, particularly in developing countries, so as to reduce the requirements for NGS or when the latter approach is not accessible.

In the present study, we evaluated the molecular alterations in *BRAF* and histone 3 genes in pediatric glial and glioneuronal tumors from our institution. We also correlated the results with clinical and histological findings in our pediatric cohort to evaluate their usefulness as diagnostic or prognostic tools in this age group.

## Materials and methods

### Ethics statement

Hospital's ethics committee, *Comité de Ética en Investigación*, has reviewed and approved this study (CEI N° 19.19), which is in accordance with the human experimentation guidelines of our institution and with the Helsinki Declaration of 1975, as revised in 1983. Biopsy samples from patients with CNS tumors were anonymized prior to this study. A written informed consent was obtained from all the patient's parents or tutors.

### Patients and samples

One hundred and two newly diagnosed glial or glioneuronal tumor patients who had undergone surgery, received treatment and follow-up at Ricardo Gutiérrez Children Hospital between 2016 and 2020 were enrolled. All were pediatric cases under 18 years of age, 50% females, and with a median age of 120 months (range 3–204 months). Of these, 82 were LGG with a median age of 120 months (range 3–204 months) and 20 were HGG with a median age of 132 months (range 12–204 months). Cases with syndromic association, especially Neurofibromatosis 1, were excluded.

Intrasurgical biopsies were obtained for diagnostic purposes. A small fragment was spread on a slide and fixed with Carnoy's solution. The remaining fragment was formalin-fixed and paraffin embedded (FFPE). Histological sections were blindly evaluated by two independent pathologists (SLC and EDM). After histological and molecular analysis, cases were classified in grades according to the WHO 2016 classification, LGG (G I and II) and HGG (G III and IV) [2].

### Immunohistochemical analysis

Four micrometer thick FFPE tissue slides were stained in an automated BenchMark XT instrument (Ventana-Roche, USA) with antibodies anti-GFAP (clone G-A-5, Bio SB, USA), -Synaptophysin (clone SP11, Roche, USA), -neurofilament (clone SF11, Cell Marque, USA), -Ki67 (clone 30–9, Roche, USA), -OLIG2 (clone EP112, Bio SB, USA), -H3K27M (clone ABE419,

Millipore, USA), -BRAF V600E (Clone VE1, Ventana-Roche, USA), -ATRX (clone BSB-108, Bio SB, USA) and -Tri-Methyl-Histone H3 (Lys27) (clone C36B11, Cell Signaling, USA). Antigen retrieval was performed with cell conditioning solution 1 or 2 (Ventana-Roche, USA), as required by each antibody, and amplification was performed with UltraView Universal DAB (Ventana-Roche, USA). Each slide contained a previously characterized positive tissue section for each assayed antibody, as a positive control.

Existence of a mutated ATRX or Tri-Methyl-Histone H3 (Lys27) was interpreted as: retained expression (wild-type gene) when nuclear staining was observed in tumor cells or, loss-of-expression (mutated gene) when no staining was observed in tumor cells but was still reactive in native ones, like endothelial cells and/or inflammatory infiltrating cells, which served as a sample specific internal control.

## Fluorescent in Situ Hybridization (FISH)

Carnoy's solution-fixed spread (CSFS) material slides were assayed by FISH in all LGG cases for KIAA1549-BRAF gene fusion. Additionally, those LGG positive cases for BRAF V600E were also assayed for deletions in *CDKN2A* gene. HGG were also screened for *CDKN2A* gene homozygous deletion.

Dual-color interphase FISH was performed for the detection of the KIAA-BRAF fusion. Probes complementary to the duplicated site of BRAF gene were labeled with rhodamine, and probes for KIAA149 were labeled with fluorescein isothiocyanate (FITC). Locus specific probe complementary to *CDKN2A* gene was labeled with rhodamine, and the centromere of chromosome 9 was labeled with FITC. All probes, BRAF1/KIAA1549 and CDKN2A/CEN9p21 LIVe Dual Color FISH Probes, were purchased from LEXEL Laboratory (LEXEL, Argentina) and FISH was performed following manufacturer's protocols.

Briefly, CSFS were rinsed once with phosphate saline buffer (PBS) 1x for 5 minutes and tissue sections were digested with porcine stomach mucosa pepsin (Sigma, USA) solution (0,1 gr/ml) for 5 minutes at 37˚C, and rinsed twice in PBS. Spreads were dehydrated with a 70%/100% ethanol series. Probes (10 μl) were then applied and the sections were covered with a coverslip and edges were sealed with rubber cement. Spreads were denatured at 75˚C for 5 minutes, followed by incubation at 37˚C overnight in a ThermoBrite incubator (Abbott Molecular, USA). Slides were then washed with 2x citrate saline solution (CSS)/0.3% NP-40 at 72˚C for 1–2 minutes, followed by a second rinse with 2x CSS at room temperature for 2 minutes. Nuclei were counterstained with 4',6-diamidino-2-phenylindole (DAPI), and fluorescence was visualized in an AxioScopeA1 epi-fluorescence microscope (Carl-Zeiss, Germany) with filter set 01 for DAPI, filter set 09 for FITC, filter set 20 for rhodamine and the dual-pass filter set 25 for FITC/rhodamine (Carl-Zeiss, Germany). A minimum of 100 tumor nuclei were evaluated and specimens were considered positive for the KIAA1549:BRAF fusion when doublets of red and green signals could be observed overlapping each other, as opposed to signals separated by 1 signal diameter, which characterizes wild type alleles. Additionally, tandem duplications of the red signals were also evaluated as a confirmation of BRAF duplication. Images were acquired with an Axiocam 503 color dedicated camera (Carl-Zeiss, Germany).

## Molecular analysis

Material from FFPE blocks was amplified by PCR and directly sequenced by Sanger method.

**Nucleic acid purification and PCR amplification.** Total DNA was purified from up to four 20 μm thick FFPE tissue sections with High Pure PCR Template Preparation Kit, (Roche, Germany), according to the manufacturer's instructions. Purified DNA was quantified in a NanoDrop One instrument (Thermo Scientific, USA) and the relation between absorbance at

**Table 1. Primers and PCR amplification conditions.**

| Amplified gene | Primers 5'- 3' | Annealing temperature (°C) | Fragment length (bp) |
|---|---|---|---|
| BRAF | Braf-fwd: TCATAATGCTTGCTCTGATAGGA | 55 | 224 |
| | Braf-rev: GGCCAAAAATTTAATCAGTGGA | | |
| H3F3A | H3.3-fwd: TGCTGGTAGGTAAGTAAGGAG | 50 | 312 |
| | H3.3-rev: TTTCCTGTTATCCATCTTTTTGTT | | |
| HISTH3B | H3.1-fwd: TTGGTGGTCTGACTCTATAAAAGAA | 55 | 214 |
| | H3.1-rev: CGGTAACGGTGAGGCTTTT | | |

260 and 280 nm was assessed; samples with a 260/280 relation above 1,8 were considered adequate as PCR templates.

All amplifications were performed in 25 μl final volume with Platinum Taq DNA Polymerase (Invitrogen, USA) following manufacturer's instructions. Specific primers and annealing temperatures for each amplified region are described in Table 1. Amplifications were carried out in a Veriti thermal cycler (Applied Biosystems, USA).

**Automated Sanger sequencing.** PCR products were subjected to electrophoresis in 2% agarose gel, specific DNA bands were cut from the gel with a sterile scalpel and purified with a QIAEXII gel extraction kit (Qiagen, Germany). These purified PCR products were directly sequenced using the BigDye Terminator 3.1 kit (Applied Biosystems, USA) in an automated 3500 series Genetic Analyzer (Applied Biosystems, USA). At least two independent sequencing reactions (sense and anti-sense) were performed with the same PCR primers to confirm each sequence. Sequencing fasta files were analyzed in Bioedit v7.0 [17].

## Statistical analysis

Chi-squared test ($\chi$2-test) was used to examine the correlation between molecular markers and clinical parameters. Overall survival (OS) was defined as time from diagnosis to death or the time to last follow-up for periods $\geq$ than 60 months. Progression-free survival (PFS) was defined as the time from tumor diagnosis to recurrence or progression. Periods $\geq$ than 60 months were considered for the progression-free category. Survival curves were performed by Kaplan–Meier method. All tests were performed with GraphPad Prism v5.01 (GraphPad Software, USA) and $P$ values $< 0.05$ were considered significant.

## Results

In the course of 4 years, 102 pediatric patients with diverse CNS tumors were diagnosed and treated at our institution. All diagnostics were performed at the Pathology Division following morphologic evaluation with the complementation of anti-synaptophysin and anti-neurofilament antibodies for neuronal lineage, anti-GFAP and anti-OLIG2 for glial lineage and anti-Ki67 for assessing proliferation index. As a complement for diffuse astrocytoma diagnosis, anti-ATRX was used. Following WHO classification, 82 cases were LGG and 20 were HGG. Among the LGG, 64 were pilocytic astrocytomas, 9 gangliogliomas, 4 pleomorphic xanthoastrocytoma and 5 GII diffuse astrocytomas. Among HGG, 9 were GIV diffuse midline glioma H3 K27M-mutant, 4 GIV glioblastomas, 4 GIII diffuse anaplastic astrocytoma, 1 GIII anaplastic oligodendroglioma, 1 GIII anaplastic pleomorphic xanthoastrocytoma, and the final one was a GIII anaplastic ganglioglioma. Regarding anatomical location of the low grade tumors, 24 were located in the midline, 35 in the cerebellum and 23 in the brain hemispheres. High grade tumors were located in the midline, 9 cases, and the remaining 11 were located, 10 in

brain hemispheres and 1 case in the cerebellum. Demographic, clinical and histological data can be found in Table 2. The study's minimal underlying data set is available in S1 Table.

## Detection of the KIAA-BRAF fusion

In order to visually evaluate the *KIAA1549-BRAF* fusion, FISH technique, with dual-color fusion probes, was applied to 78/82 LGG, with the exception of the already characterized pleomorphic xanthoastrocytomas, since the fusion has not been reported for this entity. One diffuse astrocytoma and 43 pilocytic astrocytomas were positive for the *KIAA1549-BRAF* fusion (Fig 1A and 1B). Among those positive cases, 27/44 (61.3%) were located in the cerebellum, 13/44 (29.5%) in the midline and the remaining 4/44 (9.2%) in the brain hemispheres. On the other hand, all 9 gangliogliomas and the remaining 25 LGG cases were negative for the *KIAA1549-BRAF* fusion.

Regarding *KIAA1549-BRAF* fusion in pilocytic astrocytomas and diffuse astrocytomas, there was an association between the fusion and its anatomical location to the cerebellum (Chi square test; $P = 0.04$). No association was found when analyzing different age groups (10 < years old $\geq$ 10, Fisher's exact test; $P > 0.05$). Kapplan-Meier analysis demonstrated that *KIAA1549-BRAF* fusion was not associated with PFS or with OS in LGG (Fig 1C and 1D).

## BRAF V600E molecular detection

The molecular assessment of the BRAF V600E mutation by Sanger sequencing was performed in 82 LGG, 3 of which did not amplify by PCR, and 20 HGG. Among the LGG, 10 cases were

**Table 2. Demographic, clinical, and histological patient features.**

| Cohort clinical data | LGG (n = 82) | HGG (n = 20) |
|---|---|---|
| **Sex** | | |
| Male | 40 | 10 |
| Female | 42 | 10 |
| **Age at diagnosis (months)** | | |
| Median | 120 | 132 |
| Range | 3–204 | 12–204 |
| **Tumor site** | | |
| Hypothalamic/chiasmatic | 6 | 0 |
| Brainstem | 10 | 6 |
| Thalamic | 5 | 2 |
| Spinal cord | 3 | 1 |
| Cerebellum | 35 | 1 |
| Hemisferic | 23 | 10 |
| **Pathology** | | |
| Pilocytic astrocytoma | 64 | |
| Ganglioglioma | 9 | |
| Pleomorphic xanthoastrocytoma | 4 | |
| Diffuse astrocytoma | 5 | |
| Glioblastoma | | 4 |
| Anaplastic oligodendroglioma | | 1 |
| Diffuse midline glioma H3 K27M-mutant | | 9 |
| Diffuse anaplastic astrocytoma | | 4 |
| Anaplastic pleomorphic xanthoastrocytoma | | 1 |
| Anaplastic ganglioglioma | | 1 |

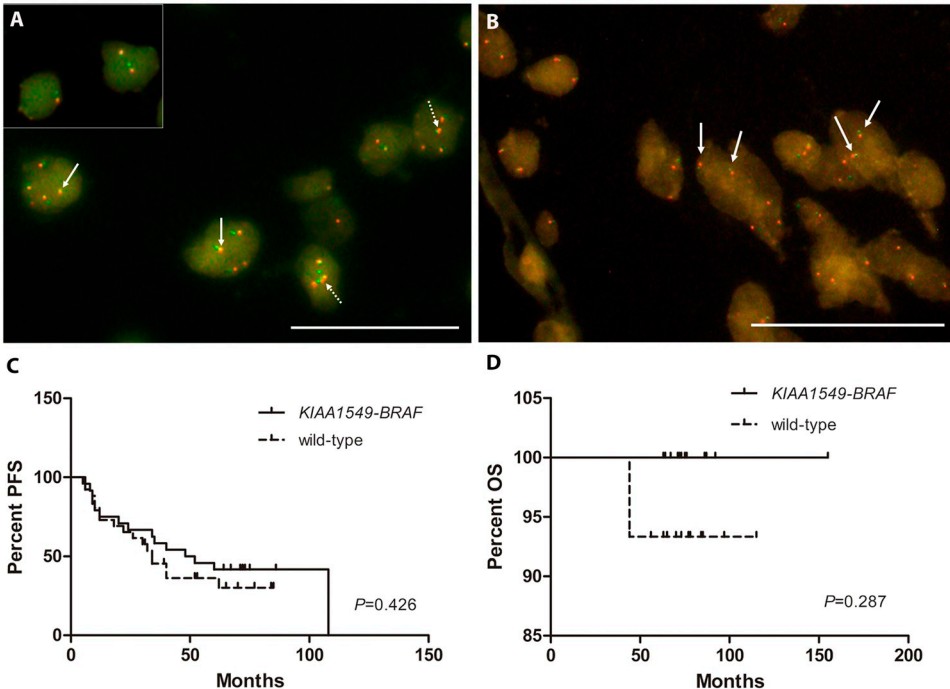

**Fig 1. *KIAA1549-BRAF* gene fusion analysis. A)** Representative pilocytic astrocytoma case showing positive fusion in interphase nuclei by FISH. *BRAF* probe in red, *KIAA1549* probe in green and overlapped probes in yellow (continuous arrows). *BRAF* gene duplication is indicated by dotted arrows. Insert depicts another fusion-positive case without *BRAF* gene duplication. Scale bar represents 50 μm at 1000x magnification. **B)** Representative image for a *KIAA1549-BRAF* gene fusion-negative case (arrows show contiguous but separated red and green signals). *BRAF* probe in red, *KIAA1549* probe in green. Scale bar represents 50 μm at 1000x magnification. **C)** Kaplan-Meyer progression free survival analysis in LGG. **D)** Kaplan-Meyer overall survival analysis in LGG.

positive for BRAF V600E mutation (4 gangliogliomas, 3 pilocytic astrocytoma, 2 pleomorphic xanthoastrocytoma and 1 diffuse astrocytoma). All cases were heterozygous for the mutation (Fig 2A). As expected, those cases positive for *KIAA1549-BRAF* fusion were negative for BRAF V600E mutation. Moreover, those 10 cases positive for the mutation were negative for *CDKN2A* deletion and loss of ATRX. Only one case with BRAF V600E mutation, was also p53 mutant; however, to date this patient achieved a 36 months of complete remission period and is still in follow-up. Regarding anatomical location, tumors positive for BRAF V600E mutation were located at the midline 3/10 (30%) and in the brain hemispheres 7/10 (70%), while no mutated cases were present in the cerebellum, which suggests that BRAF V600E mutation does not occur in LGG in this anatomical location (Fisher's exact test; $P = 0.0041$). No association between the mutation and patient´s age at onset was observed (Fisher's exact test; $P>0.05$).

On the other hand, among HGG, just 1 glioblastoma located in the brain hemisphere was positive for the mutation. Only in HGG cases, BRAF V600E mutation was also assessed by immunohistochemistry (IHC) with an antibody directed against this specific mutation. In all these cases results obtained by Sanger sequencing and IHC correlated with each other (Fig 2B). Moreover, all HGG cases were negative for *CDKN2A* deletion.

Regarding survival outcome in LGG, Kaplan-Meier analysis demonstrated that BRAF V600E mutation was associated with a shorter OS (Log-rank (Mantel-Cox) Test, $P = 0.0082$) (Fig 2C), but not with a decreased PFS (Log-rank (Mantel-Cox) Test, $P = 0.14$) (Fig 2D). However, when only considering pilocytic astrocytoma, the BRAF V600E mutation was associated

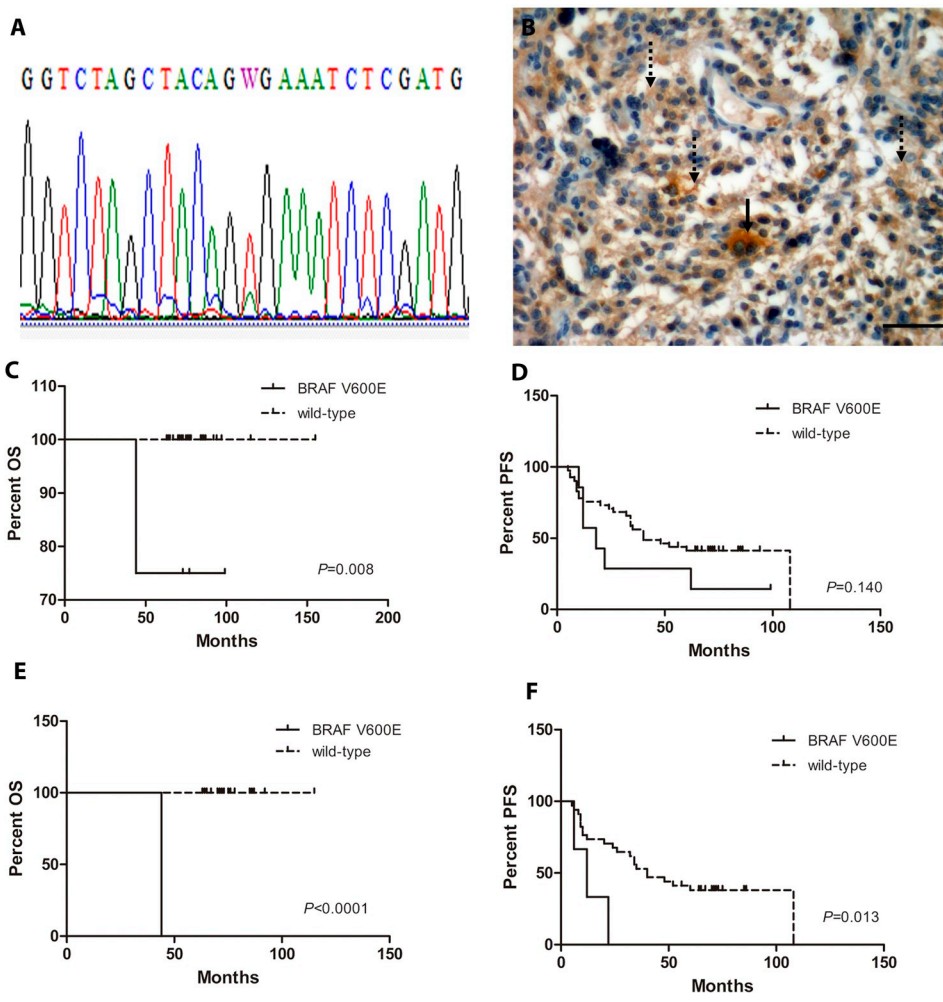

**Fig 2. BRAF V600E mutation analysis. A)** Representative Sanger sequencing chromatogram showing double peaks in c.1799A>T (p.V600E), indicating a BRAF V600E heterozygous mutation. **B)** Representative immunohistochemistry for BRAF V600E mutation in a ganglioglioma case showing positive signal in glial processes (dotted arrows) and tumor ganglion cells (continuous arrow). Scale bar represents 50 μm at 400x magnification. **C)** Kaplan-Meyer overall survival analysis in LGG. **D)** Kaplan-Meyer progression free survival analysis in LGG. **E)** Kaplan-Meyer overall survival analysis in pilocytic astrocytoma. **F)** Kaplan-Meyer progression free survival analysis in pilocytic astrocytoma.

with a decreased OS (Log-rank (Mantel-Cox) Test, $P<0.0001$) (Fig 2E) and PFS (Log-rank (Mantel-Cox) Test, $P = 0.0135$) (Fig 2F). On the other hand, BRAF V600E mutation was not associated with outcome in patients with HGG.

## Molecular characterization of histone 3

The molecular assessment of the H3K27M mutation by Sanger sequencing was performed in 24 LGG of the midline. From these, all 22 amplifiable cases were wild type for *H3F3A*, *HIST1H3B* and *HIST1H3C* genes.

Additionally, 20 HGG were also assessed for mutations in histone genes by Sanger sequencing, where 2 cases did not amplify. From the remaining 18, all 9 HGG of the midline were positive for H3K27M mutation (8 in H3.3 and 1 in H3.1 isoforms) while 2 glioblastomas harbored the H3G34R mutation; the remaining 2 glioblastomas, 4 diffuse astrocytomas and 1 anaplastic oligodendroglioma were wild type for the histone genes (Fig 3A–3C). It is important to

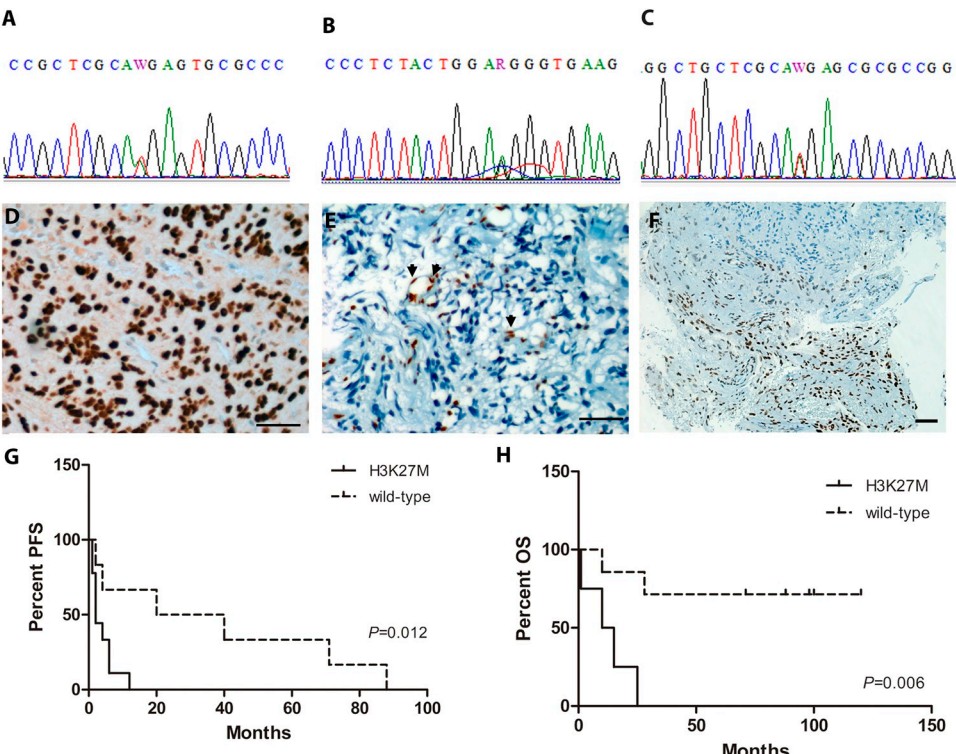

**Fig 3. Histone 3 mutation analysis. A)** Representative Sanger sequencing chromatogram showing double peaks in c.83A>T (p.K27M), indicating a H3K27M heterozygous mutation in isoform H3.3. **B)** Representative Sanger sequencing chromatogram showing double peaks in c.103A>G (p.G34R), indicating a H3G34R heterozygous mutation in isoform H3.3. **C)** Representative Sanger sequencing chromatogram showing double peaks in A>T (p. K27M), indicating a H3K27M heterozygous mutation in isoform H3.1. **D)** Representative immunohistochemistry for positive midline diffuse glioma H3K27M mutant case. Scale bar represents 50 μm at 400x magnification. **E)** Representative immunohistochemistry for H3K27me3 in the H3.3 mutant isoform depicts a complete loss-of-staining in tumor cells, while endothelial cells retain a positive staining status (arrows). Scale bar represents 50 μm at 400x magnification. **F)** Immunohistochemistry for H3K27me3 in the H3.1 mutant isoform case depicts a heterogeneous mosaic loss-of-staining. Scale bar represents 50 μm at 200x magnification. **G)** Kaplan-Meyer progression free survival analysis in HGG. **H)** Kaplan-Meyer overall survival analysis in HGG.

mention that both H3G34R mutant glioblastomas were located in brain hemispheres. Seeking to correlate results from Sanger sequencing and IHC detection of the H3K27M mutation, all 9 midline diffuse glioma H3K27M mutant cases, rendered as mutated by sequencing, were assessed by IHC with a H3K27M mutation specific antibody and, independently, with an antibody specific for tri-methylated Lys 27 (Fig 3D and 3E). Absolute congruence was observed in all of these cases when comparing the three detection methods; however, it was surprising to note that the only case of H3K27M, mutated on histone isoform 3.1, showed a particular pattern of signal-loss with the trimethylated-Lys27 antibody. While all the other 8 cases, mutated on histone 3.3 isoform, showed a complete loss-of-expression with trimethylated-Lys27 antibody (Fig 3E), the 3.1 mutated isoform showed a heterogeneous or mosaic loss-of-staining (Fig 3F). As negative controls for IHC, the 2 cases with H3G34R variant in histone 3.3 isoform were also assessed with both antibodies. As expected, they were negative for the H3K27M mutation and both cases retained the tri-methylation status of Lys27. Kapplan-Meier analysis demonstrated that the H3K27M mutation was associated with decreased PFS (Log-rank (Mantel-Cox) Test, $P = 0.0124$) (Fig 3G) and OS (Log-rank (Mantel-Cox) Test, $P = 0.006$) (Fig 3H).

## Discussion

Tumors of the central nervous system are still the leading cause of childhood cancer-related deaths. Despite the many advances in surgical and adjuvant therapy, which have increased survival rates, complete resection is not usually possible for inaccessible, critically-located or infiltrative tumors, which have a worse outcome than superficial lesions. In these particular cases, the identification of specific molecular markers aids in establishing prognosis and individualize treatment strategies that could result in unnecessary sequelae in a developing child. Moreover, molecular markers now assist the histological diagnosis, enabling the classification of different subset of tumors, actually recognized by WHO [2], and providing new insights, risk stratification and treatment opportunities for almost every type of pediatric brain tumor [18, 19].

Although many other gene fusions, involving several genes, have been described as critical driver events in pediatric LGG, the *KIAA1549-BRAF* fusion is one prominent molecular marker, most frequently detected and assists in the diagnosis of pilocytic astrocytoma [20]. In our series, 43/64 (67%) pilocytic astrocytomas contained this fusion, a higher proportion of cases to that previously described [15, 20, 21]. Moreover, Kurani *et. al.* detected the fusion, by means of RT-PCR and sequencing, in only 41% of their pediatric cohort; although this discrepancy in the percentage of positive cases could be due to the difference in sensitivity of the employed techniques. Similar to that observed in other studies, regardless of the histological classification, we disclosed a significant association between *KIAA1549-BRAF* gene fusion and its anatomical location to the cerebellum for LGG [21]. While we found the fusion in 61.3% of LGG, others detected it ranging from 59.1 to 89.7% of LGG [14, 15, 20, 22].

Regarding the prognostic value of the fusion and its association with OS and PFS, some controversy has been reported in the literature. While Yang et. al. described a better PFS and OS for *KIAA1549-BRAF* gene fusion in LGG, the authors also state that this was not an independent prognostic factor in the multivariate analysis, which suggests that this marker could be influenced by another factor [15]. In particular, the series described by Yang et. al. contained a high proportion of pilocytic astrocytomas, which intrinsically have a better prognosis than other LGG, and could have influenced their observations. Conversely, in our series the gene fusion was not associated with a modified outcome, either in LGG as a whole, nor in pilocytic astrocytoma as similarly described by Faulkner et. al. [23] and by Penman et. al. in a review article [24]. Given the presence of the *KIAA1549-BRAF* gene fusion in over 67% of our pilocytic astrocytoma cases, it could be more of a useful diagnostic tool for this entity, than a prognosis related marker.

Similarly, the BRAF V600E mutation is also a useful molecular marker to estimate evolution and prognosis. Moreover, a recent report described the regression of BRAF mutated tumors in response to the BRAF inhibitor dabrafenib [25], an observation of particular importance in not resectable tumors. Overall, in LGG, we detected the mutation in 12.6% of the cases; however this proportion rises to 50% when only considering gangliogliomas, a proportion in accordance to that described by previous studies which found the BRAF V600E mutation in between 20–60% of cases [9, 26–30]. Particularly in gangliogliomas, which could compromise deep structures of the brain, preventing complete surgical resection and achieving adequate oncological control, BRAF V600E mutation could represent a druggable target for specific inhibitors such as vemurafenib or dabrafenib. Although these treatment options are still in trial and lack a pediatric formal indication, the International Consortium on Low Grade Glioma (ICLGG) of the International Society of Pediatric Oncology (SIOP) recommend their use in these cases [31]. In pilocytic astrocytoma, we detected the mutation in 3/62 (4.8%) of cases, again in accordance to previous reports using direct Sanger sequencing [21, 32].

These results further prove that *KIAA1549-BRAF* gene fusion was the most common *BRAF* alteration in pilocytic astrocytoma, while BRAF V600E was the most predominant in ganglio-gliomas. In all cases, BRAF V600E was assessed by IHC (VE1 clone) and by Sanger sequencing. Results obtained by both methods were congruent with each other, which suppose an advantage for pathology labs without access to a sequencing facility.

In our hands, when only considering pilocytic astrocytoma cases, the BRAF V600E was associated with a worse outcome for both PFS and OS. However, when considering the entire series of LGG, BRAF V600E was only associated with a worse OS, but not with PFS; a fact that could be due to the inclusion of only the subgroup of LGG that fulfilled the 5 year period of follow-up. Controversy still arises when considering BRAF V600E as a prognostic factor for pediatric LGG. While some researchers reported it to be a useful prognostic marker for PFS and OS [15], others reported it to be a useful prognostic factor for PFS, but not OS [11]. Our results are contradicting with this latter study since we reported BRAF V600E to be associated with a worse OS but not with PFS, something that could be due to the differences in age, tumor location and extent of surgical resection between both series of patients. Moreover, and to the best of our knowledge, no other report has explored its prognostic association with PFS or OS exclusively in pediatric pilocytic astrocytomas. Altogether, alterations in *BRAF* gene do not only constitute a valuable diagnosis tool and prognostic factor for some cases of LGG, particularly pilocytic astrocytoma, but also represents an interesting candidate for targeted therapy aimed at reducing the constitutively activated MAPK pathway.

Regarding the contribution of BRAF V600E mutation as a prognostic factor in HGG, a recent study on pediatric HGG identified the mutation in 5/42 cases, three of which were LGG that progressed to HGG [33]. From this observation, the authors suggested that LGG should be assessed for BRAF V600E mutation and those mutant tumors should be closely followed-up. In our series we detected the mutation in only one case of HGG, which rapidly progressed and died within 28 months from diagnosis.

The molecular assessment of the H3K27M mutation by Sanger sequencing was determined in 22 LGG of the midline, all of which were wild type for *H3F3A*, *HIST1H3B* and *HIST1H3C* genes. The diagnostic reasoning behind only including gliomas of the midline, among LGG, for H3K27M assessment was its reported prevalence in this anatomic location together with a disfavorable outcome [2]. Moreover, according to WHO guidelines, those cases histologically characterized as LGG of the midline that contain the H3K27M mutation are reclassified as high grade diffuse midline glioma H3K27M mutant. However, recent reports described that these cases, with low grade histology and H3K27M mutation, may not have such an unfavorable outcome as the high grade diffuse midline glioma H3K27M mutant, or have such a slow progression rate that exceeds the 5 year follow-up periods. Hence, the assessment of H3K27M mutations should be performed in all pediatric midline gliomas independent of their histologic appearance or grade [34, 35]. In line with WHO guidelines, in our series, the only LGG of the midline that harbored the HK27M variant in isoforms H3.3 (coded by *H3F3A* gene), was reclassified as a high grade diffuse midline glioma H3K27M mutant. However, Yang et. al. reported the H3K27M mutation in only 6.4% LGG, of which 50% were in the midline [15]; probably applying a different criteria for their classification.

Concerning HGG, mutations in genes coding for histone H3 (*H3F3A*, *HIST1H3B* and *HIST1H3C*) assessed by Sanger sequencing revealed that 61% of the cases contained a mutation in one of these genes, either H3K27M in 9 cases or H3G34R in the remaining 2 cases. When only considering the occurrence of H3K27M mutation, 50% of our cases harbored it, a similar proportion to that previously described by Bozkurt et. al. and Huang et. al. [5, 6].

Among those cases with histone H3 mutations, 9 cases were H3K27M mutant (82%), all of which were of midline localization, while the two remaining cases harboring the H3G34R

mutation were located in the hemispheres. These latter findings are also in line with previous reports that show a predilection of H3K27M to midline locations and H3G34R to the hemispheres in pediatric HGG [6, 15, 36]. Also in line with the literature, only 1/11 HGG contained the mutation in the *HIST1H3B* gene (isoform H3.1), while the remaining 10 cases harbored the mutation in the *H3F3A* gene (isoform H3.3) [33, 37].

Additionally, we observed an absolute congruence between Sanger sequencing and the IHC results in all studied cases for H3K27M. Similar concordance was reported by Huang et. al. when assessing by sequencing a few cases within their studied series [5]. Moreover, the congruency in results was also observed when assessing the loss of tri-methylation status of Lys27 (H3K27me3) in isoform H3.3 and H3.1 by IHC. Regarding the case with H3K27M in isoform H3.1, as detected by sequencing, we described by IHC a patched pattern loss-of-signal with the H3K27me3 antibody. This was a worth mentioning observation, and a similar result was reported by Castel et. al. who described a weak staining with the H3K27me3 antibody in a H3K27M mutant in isoform H3.1 [36]. Additionally this same mosaic pattern of signal-loss was also described for H3K27me3 antibody in cases of malignant nerve sheath tumors [38]. Although there is no clear explanation for this atypical staining pattern, it could be due to the fact that H3.3 is expressed throughout the cell cycle, as well as in quiescent cells, but H3.1 and H3.2 are cell cycle regulated and deposited only during the S-phase and during DNA repair. Given differences in expression and deposition, levels of H3.1 and H3.3 protein isoforms may vary widely in different cell types, tissues, and cell-cycle moments ([7] and references herein). Finally, as previously reported, the H3K27M mutation was associated with a worse outcome, regarding both PFS and OS in HG [5, 33, 37].

Particularly in developing countries where access to high throughput sequencing is still limited, assessing these biomarkers by means of conventional molecular biology techniques is of paramount importance, since they aid in pediatric CNS tumors classification. In fact, this new molecular classification approach aims to reflect the biological immunophenotype of the tumor rather than its morphology, with the ultimate goal of achieving a risk-adapted framework and molecularly targeted therapies which augment or, in some cases, replace conventional therapy.

## Supporting information

**S1 Table. Minimal underlying data set for LGG and HGG.**
(XLSX)

## Author Contributions

**Conceptualization:** Mario Alejandro Lorenzetti, María Victoria Preciado.

**Data curation:** Sandra Lorena Colli, Nazarena Cardoso, María Cores, Mario Alejandro Lorenzetti, María Victoria Preciado.

**Formal analysis:** Nazarena Cardoso, Carla Antonella Massone, María Cores, Mercedes García Lombardi, Elena Noemí De Matteo, Mario Alejandro Lorenzetti, María Victoria Preciado.

**Funding acquisition:** María Victoria Preciado.

**Investigation:** Sandra Lorena Colli, Mario Alejandro Lorenzetti, María Victoria Preciado.

**Methodology:** Sandra Lorena Colli, Nazarena Cardoso, Carla Antonella Massone, Mario Alejandro Lorenzetti.

**Project administration:** María Victoria Preciado.

**Resources:** Elena Noemí De Matteo, María Victoria Preciado.

**Supervision:** Mario Alejandro Lorenzetti, María Victoria Preciado.

**Validation:** Carla Antonella Massone, Mercedes García Lombardi, Elena Noemí De Matteo, Mario Alejandro Lorenzetti.

**Visualization:** Mercedes García Lombardi, Elena Noemí De Matteo, Mario Alejandro Lorenzetti.

**Writing – original draft:** Mario Alejandro Lorenzetti, María Victoria Preciado.

**Writing – review & editing:** Sandra Lorena Colli, Nazarena Cardoso, Carla Antonella Massone, María Cores, Mercedes García Lombardi, Elena Noemí De Matteo, Mario Alejandro Lorenzetti, María Victoria Preciado.

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
