## [Decision Letter · Decision Letter 0]

25 Feb 2022

PONE-D-21-37332Molecular alterations in the integrated diagnosis of pediatric glial and glioneuronal tumors: a single center experiencePLOS ONE

Dear Dr. Preciado,

Thank you for submitting your manuscript to PLOS ONE. After careful consideration, we feel that it has merit but does not fully meet PLOS ONE’s publication criteria as it currently stands. Therefore, we invite you to submit a revised version of the manuscript that addresses the points raised during the review process. Please submit your revised manuscript by Apr 11 2022 11:59PM. If you will need more time than this to complete your revisions, please reply to this message or contact the journal office at plosone@plos.org. Please include the following items when submitting your revised manuscript:A rebuttal letter that responds to each point raised by the academic editor and reviewer(s). You should upload this letter as a separate file labeled 'Response to Reviewers'.A marked-up copy of your manuscript that highlights changes made to the original version. You should upload this as a separate file labeled 'Revised Manuscript with Track Changes'.An unmarked version of your revised paper without tracked changes. You should upload this as a separate file labeled 'Manuscript'.

We look forward to receiving your revised manuscript.

Kind regards,

Kevin Camphausen

Academic Editor

PLOS ONE

Journal Requirements:

2. Thank you for stating in your Funding Statement: "This study was supported in part by a grant from the National Institute of Cancer (INC) of the National Ministry of Health (Res 83/2020) and National Research Council (CONICET) PUE 2018 nº0058. E.N.D.M, M.A.L, and M.V.P are members of the CONICET Research Career Program. C.A.M. was supported by a fellowship from INC and N.C. by a fellowship from Buenos Aires City Ministry of Health.

Reviewers' comments:

Reviewer's Responses to Questions

**Comments to the Author**

1. Is the manuscript technically sound, and do the data support the conclusions?

Reviewer #1: Yes

Reviewer #2: Yes

2. Has the statistical analysis been performed appropriately and rigorously? 

Reviewer #1: Yes

Reviewer #2: Yes

3. Have the authors made all data underlying the findings in their manuscript fully available?

Reviewer #1: Yes

Reviewer #2: Yes

4. Is the manuscript presented in an intelligible fashion and written in standard English?

Reviewer #1: Yes

Reviewer #2: Yes

5. Review Comments to the Author

Reviewer #1: The manuscript presents data on gene mutation (PCR and sequencing), fusion and deletion (FISH) and protein expression (IHC) of BRAF and Histone H3 in gliomas and correlated with treatment outcome. Molecular markers for diagnosis and prognostication in cancers are still debated. The data may contribute further clarity to this area.

Suggestions for improvement:

1. Abstract, last few lines: ‘for diagnostic centers without access to NGS…’ as well as elsewhere in manuscript: This statement may alternatively be written as : These potential diagnostic/prognostic markers may be suitable as further screening tests to reduce requirement on NGS, which is not available in all laboratories.

Abstract, line 7, “Study design: FISH, IHQ …” IHQ should be replaced with IHC here and elsewhere in the manuscript.

Page 6, Immunohistochemical analysis, line 9, “…performed with CC1 or CC2 solutions…”: Not clear what are the CC1 and CC2 solutions.

Tables 1 and 2 were not found.

This manuscript could benefit from a table contrasting the distribution in molecular changes and treatment outcome of LGG and HGG.

Reviewer #2: In this manuscript Preciadoet et al, provide an approach comparing different molecular tools(FISH,IHQ, and NGS), to determine the prognostic and diagnostic value, in pediatric LGG and HGG.Overall there approach is logistic and they provided good evidence to support the conclusions they make. They make an excellent point of the usefulness of these techniques, in places where NGS is not readily available. The image quality in figure 1 a and 1 b is not the best and higher resolution images would benefit the reader, and better support the data. The English is mostly grammatically correct, however the writing should be reviewed as many sentences need to be re-written.

6. PLOS authors have the option to publish the peer review history of their article (what does this mean?). If published, this will include your full peer review and any attached files.

Reviewer #1: No

Reviewer #2: No

---

## [Author Response · Author response to Decision Letter 0]

8 Mar 2022

-- Reviewer 1 --

The manuscript presents data on gene mutation (PCR and sequencing), fusion and deletion (FISH) and protein expression (IHC) of BRAF and Histone H3 in gliomas and correlated with treatment outcome. Molecular markers for diagnosis and prognostication in cancers are still debated. The data may contribute further clarity to this area.

Response: We thank Reviewer 1 for his positive comments

Suggestions for improvement:

1. Abstract, last few lines: ‘for diagnostic centers without access to NGS…’ as well as elsewhere in manuscript: This statement may alternatively be written as: These potential diagnostic/prognostic markers may be suitable as further screening tests to reduce requirement on NGS, which is not available in all laboratories.

Abstract, line 7, “Study design: FISH, IHQ …” IHQ should be replaced with IHC here and elsewhere in the manuscript.

Response: According to the reviewer´s suggestions, we replaced the above mentioned sentence in the Abstract. All IHQ misspellings were corrected to IHC.

Page 6, Immunohistochemical analysis, line 9, “…performed with CC1 or CC2 solutions…”: Not clear what are the CC1 and CC2 solutions.

Response: CC1 and CC2 are the abbreviations for Roche´s commercial solutions for cell conditioning. Since this was the only time in the manuscript that we used this abbreviation, we opted for writing it in full.

Tables 1 and 2 were not found.

Response: Following the journal´s instructions, Tables 1 and 2 were uploaded as separate files and not embedded in the manuscript body. In the system-produced merged pdf file, links to tables were present at the end of the document.

This manuscript could benefit from a table contrasting the distribution in molecular changes and treatment outcome of LGG and HGG.

Response: A Supplementary table, with two separate tabs for LGG and HGG, containing all detailed information regarding molecular marker status, follow-up periods and outcome was included to this revised submission.

-- Reviewer 2 --

Reviewer #2: In this manuscript Preciado et et al, provide an approach comparing different molecular tools (FISH,IHQ, and NGS), to determine the prognostic and diagnostic value, in pediatric LGG and HGG.

Overall there approach is logistic and they provided good evidence to support the conclusions they make. They make an excellent point of the usefulness of these techniques, in places where NGS is not readily available. 

Response: We appreciate Reviewer´s 2 comment on our work.

The image quality in figure 1 a and 1 b is not the best and higher resolution images would benefit the reader, and better support the data. 

Response: Reviewer 2 probably is referring to the figure in the merged pdf file automatically produced by the journal´s submission system for review purposes. The separate image file has much better resolution quality than the one appearing in the pdf file. We believe that the original file meets the journal´s quality requirements. 

The English is mostly grammatically correct, however the writing should be reviewed as many sentences need to be re-written.

Response: taking into consideration reviewer´s 2 comment, we thoroughly revised the manuscript and corrected several sentences with grammatical mistakes. These changes are visible in the active track-change version of the manuscript.

---

## [Editor Report · Decision Letter 1]

22 Mar 2022

Molecular alterations in the integrated diagnosis of pediatric glial and glioneuronal tumors: a single center experience

PONE-D-21-37332R1

Dear Dr. Preciado,

We’re pleased to inform you that your manuscript has been judged scientifically suitable for publication and will be formally accepted for publication once it meets all outstanding technical requirements.

Kind regards,

Kevin Camphausen

Academic Editor

PLOS ONE
---

## [Editor Report · Acceptance letter]

24 Mar 2022

PONE-D-21-37332R1 

Molecular alterations in the integrated diagnosis of pediatric glial and glioneuronal tumors: a single center experience 

Dear Dr. Preciado:

I'm pleased to inform you that your manuscript has been deemed suitable for publication in PLOS ONE. Congratulations! Your manuscript is now with our production department. 

Kind regards, 

on behalf of

Dr. Kevin Camphausen 

Academic Editor

PLOS ONE